Bloom evenness modulates the influence of bloom abundance on insect community structure in suburban gardens

http://orcid.org/0000-0002-2245-5677 Braatz Elizabeth Y. 1 2 ebraatz@ufl.edu
Gezon Zachariah J. 2
Rossetti Kristin 3
http://orcid.org/0000-0003-0443-1580 Maynard Lily T. 2
http://orcid.org/0000-0002-4726-6995 Bremer Jonathan S. 1
Hill Geena M. 3
Streifel Marissa A. 1
Daniels Jaret C. 3
1 Department of Entomology and Nematology, University of Florida , Gainesville, FL , USA
2 Conservation Department, Disney’s Animals, Science and Environment , Lake Buena Vista, FL , USA
3 McGuire Center for Lepidoptera and Biodiversity, Florida Museum of Natural History , Gainesville, FL , USA
Huber Dezene
Electronic publication date: 2021 Apr 22
Publication date: 2021
Volume: 9
Electronic Location ID: e11132
Received 2020 Aug 24; Accepted 2021 Mar 1
Copyright: © 2021 Braatz et al.
Copyright year: 2021
Copyright holder: Braatz et al.
License: This is an open access article distributed under the terms of the Creative Commons Attribution License, which permits unrestricted use, distribution, reproduction and adaptation in any medium and for any purpose provided that it is properly attributed. For attribution, the original author(s), title, publication source (PeerJ) and either DOI or URL of the article must be cited.
License URL: https://creativecommons.org/licenses/by/4.0/

Keywords: Pollinators, Conservation, Evenness, Biodiversity, Floral abundance, Urban ecology, Urban habitat, Plant richness, Insect abundance, Insect richness

Funding: Florida Wildflower Foundation (UF) 00084425 This research was funded by a grant from the Florida Wildflower Foundation (UF contract No. 00084425). The funders had no role in study design, data collection and analysis, decision to publish, or preparation of the manuscript.

==============================
As land use change drives global insect declines, the value of enhancing habitat in urban and suburban landscapes has become increasingly important for flower-visiting insects. In order to help identify best landscaping practices, we conducted plant surveys and insect bowl-trap surveys in 34 suburban yards for 21 months in Gainesville, FL, USA, which resulted in 274 paired days of plant and insect survey data. We assessed the impact of nearest greenspace size, distance to greenspace, yard area, plant richness, plant type, bloom abundance, bloom richness and bloom evenness on insect abundance and richness. Our samples include 34,972 insects captured, 485,827 blooms counted and 774 species of plants recorded. We found that bloom evenness had a modulating effect on bloom abundance—a more even sample of the same number of blooms would have a disproportionately greater positive impact on flower visitor richness, insect richness and insect abundance. Bloom abundance was also highly significant and positively associated with flower visitor abundance, but nearest greenspace size, distance to greenspace, plant type (native vs. non-native vs. Florida Friendly), and yard area were not found to be important factors. Plant richness was a highly significant factor, but its effect size was very small.

Introduction

Anthropogenic-driven land use and land cover changes are considered among the primary threats to biodiversity and the fundamental productivity and sustainability of ecological systems (Cardoso et al., 2020; Goulson, 2019; Van Klink et al., 2020). Significant biodiversity degradation is often most pronounced in urban areas due to the overall intensity of land modification leading to high levels of direct habitat loss and increased habitat fragmentation (Piano et al., 2020; Sánchez-Bayo & Wyckhuys, 2019). This in turn is commonly associated with the loss of landscape connectivity, changes in species composition, increased biotic homogenization, and the erosion of essential ecosystem services as well as innumerable other stressors and drivers that may negatively affect wildlife. These stressors are reflected in global insect populations, with one recent metastudy finding that terrestrial insect populations have declined by 24% in the last 30 years (Van Klink et al., 2020).

A driving force behind land cover change is urbanization. Today, more than half of the world’s people live in urban areas. As global population growth continues to accelerate, some 2.5 billion more people are projected to be added to cities and urban centers by 2050 (United Nations Department of Economic & Social Affairs, 2019). In the United States, over 80% of the densification of humans and human activities is concentrated in cities, with U.S. developed land area expected to increase by 40–90% in some locations (Boustan, Bunten & Hearey, 2013; Sleeter et al., 2017).

While conservation efforts and research have historically focused on more intact, high value natural areas, the utility, role and importance of urban green spaces for biodiversity conservation are increasingly becoming recognized (Rupprecht et al., 2015). Concurrent with urban population growth in the United States, the residential lawn landscape has grown to dominate the suburban and urban landscape. Within this residential landscape, lawns dominate and occupy nearly 40% of the total estimated 163,812 km2 of turfgrass that covers approximately 1.9% of the total area of the continental U.S. (Milesi et al., 2005). The non-turfgrass landscaped portion of a typical yard, which includes trees, shrubs and ornamental plantings, makes up only about 18% of the total landscape (U.S. Census Bureau, 2009). Despite this potential limitation, human-dominated environments can nonetheless be floristically diverse and harbor complex vegetation structure (Sandström, Angelstam & Mikusiński, 2006; Beninde, Veith & Hochkirch, 2015; Threlfall et al., 2016).

Plant community composition, including the fine scale habitat heterogeneity often seen in urban gardens, has been shown to influence insect community assembly. This has been particularly well-demonstrated for insect groups, with many examples of its predictive or positive correlations (Adams et al., 2020; Baldock et al., 2015, 2019; Theodorou et al., 2017). This close linkage has important ramifications for the development of appropriate conservation strategies to help maintain, increase or restore insect biodiversity in a variety of managed or degraded landscapes, especially with the growing evidence of accelerated insect declines (Potts et al., 2010; Forister, Pelton & Black, 2019; Van Klink et al., 2020; Mathiasson & Rehan, 2020). Diverse insect communities are important to ensure the provision of key ecosystem services (i.e. wildlife nutrition, decomposition, natural pest control, and pollination) and promote ecosystem resilience in the face of continued environmental change (Elmqvist et al., 2003).

Understanding which variables within an urban landscape can most impact insect populations is important for conserving insect diversity in home gardens and yards, municipal parks and greenspaces, and other built environments. Efforts to increase insect pollinator and other arthropod assemblages in urban landscapes have often focused on enhancing plant diversity (Campbell et al., 2019), sustained flowering (Williams et al., 2015), floral abundance (Campbell, Hanula & Waldrop, 2007; Campbell et al., 2019), and nativity (Salisbury et al., 2015), or various combinations of these aspects (Vrdoljak, Samways & Simaika, 2016). Nonetheless, knowledge gaps remain regarding best management practices for insect communities in urban landscapes. The built environment presents landscape architects, designers, and home gardeners with innumerable plant and design choices while simultaneously conferring constraints related to economics, space, community use, native plant commercial availability and local covenants, restrictions or ordinances.

In this study, we explored how choices gardeners made in suburban landscape plant community composition influenced the abundance and richness of insects, particularly those known to visit flowers. Our broad hypothesis was: Suburban gardens and yards are habitats for insects. Plant composition in those habitats affects recruitment/presence of wildlife, including insects and pollinators.

With the following sub-hypotheses:Insect and flower visitor richness (number of families) and abundance will increase as the number of species of plants increases.

For maximum insect abundance and richness, native plants are best, followed by Florida Friendly plants. Non-native plants are the least beneficial for maximizing insect abundance and richness.

Insect and flower visitor richness and abundance will increase as the number of blooms increase.

The ‘Florida Friendly’ designation in sub-hypothesis 2 was based on the Florida-Friendly Landscaping™ (FFL) Program, which was enacted by the Florida Legislature in 2009 in partnership with University of Florida/IFAS Extension Service, the Florida Department of Environmental Protection, and the five Water Management Districts of Florida (Momol & Tolbert, 2019). This program encourages homeowners to think about their yards’ conservation impact and to select plants that have minimal negative impact on the environment. It uses nine criteria centered around watering, pesticides, fertilizers, wildlife habitat, and other issues identified as important through UF/IFAS research efforts (UF/IFAS Florida-Friendly Landscaping Program, n.d.). UF/IFAS has assessed plants as potentially Florida Friendly by using literature-based risk-assessment tools and making regular updates to their recommendations as needed.

To answer these questions, we conducted plant and insect biodiversity surveys at 34 separate garden sites in North-Central Florida for two years. Since the gardens were privately owned, we did not manipulate any variables within the sites. We focused on these questions because they involved variables that interested landowners could potentially control in suburban landscapes.

Materials and Methods

Study sites

We conducted field portions of study in Alachua County in the North Central region of Florida within and surrounding the city of Gainesville. It is classified as zone 9a on the USDA plant hardiness zone map and as humid subtropical (Cfa) according to the Köppen–Geiger climate classification system (Kottek et al., 2006).

We studied 34 separate residential urban gardens. Our research was survey-based and our unit of measurement was the entirety of each site’s yard. We conducted surveys at the sites over a 21-month period from March 2013 to Dec 2014. Access was granted by participating homeowners.

To identify potential study properties, we broadly disseminated a call for study participants by direct email, newsletter posts and meeting presentations targeting University of Florida IFAS Extension Agents, Master Gardeners, the Sierra Club, the Florida Native Plant Society, the Audubon Society and the Florida Trail Association. We also used the property address search function on the Alachua County Property Appraiser’s website to identify neighborhoods that were viable candidates based on the size and number of eligible parcels, and then reached out to local homeowners associations in those areas (see Supplemental Information: Initial email sent to solicit participants).

Once a homeowner replied, indicating interest and willingness to participate, we verified the parcel size from the Alachua County Property Appraiser’s website and conducted a site visit to visually assess the potential property. Homes with swimming pools (which take up a majority of the yard), locked gates, or large dogs on the property that might create access or safety issues were excluded from consideration. We only considered homes with lot sizes between 0.10 and 0.20 hectares. This was considered a conservative range of typical suburban residential lot sizes in Gainesville, FL, USA. In order to ease the selection process, we did not consider the demographics of garden owners. The homeowners of all selected properties were asked to sign a property access agreement prepared by the University of Florida’s Office of the Vice President and General Counsel.

We strove to include properties that varied in composition and structure, so we selected yards that varied in terms of plant richness and nativity. Some yards were more conventional, with a few primarily ornamental, non-native accent plants. Other yards were more naturalistic, with higher abundances and diversity of native and FFL designated species. Each yard varied; we did not have replications of each group along the gradient.

Residential yards were also selected along a gradient of distance to green space. Greenspace was defined as a park, preserve, conservation area/easement, botanical garden, greenway, or undeveloped land. We measured the linear distance from each property to the nearest greenspace using ArcGIS. We noted the log-transformed acreage of the nearest greenspace for each site. Size (square acres) and distance to nearest greenspace were included as separate variables in model testing.

Based on these criteria, we chose 34 residential properties from a pool of over 60 homeowners who indicated interest in participating. Although these sites were not randomly sampled, the number of samples and variance in composition and structure mimics typical suburban to urban gardens around North Central Florida. Once a property was selected, all participants signed a legal release allowing researchers to access their yard as needed. This release detailed the goals and duration of the research study, what researchers would be doing, which types of data would be collected, the frequency and duration of visits, and informed homeowners that they could withdraw their permission/participation at any time.

Vegetation survey protocol

We conducted two different vegetation surveys for each residential yard: a comprehensive vegetation survey and a flowering plant total bloom count. Exact dates of surveying varied, depending on scheduling and weather. Due to the subtropical climate of our study location, vegetation sampling could be viably conducted on a year-round basis.

We conducted a comprehensive vegetation survey of each residential yard once per year in January 2013 and February 2014. For this, we systematically recorded every plant species in the front and back yard (except for one property where the homeowner elected to only allow their front yard to be surveyed). All plants were identified using floral morphology or non-floral characteristics (e.g., leaf morphology) if the species was not in flower at the time of the survey. Because of the subtropical climate in our study system, most plants were present during these months, though not necessarily blooming. Due to study limitations (labor constraints, minimizing time spent in homeowner backyards) and the logistical considerations of quantifying plant counts for ground covers and vines, we focused our data collection on presence/absence of plant species. Our comprehensive vegetation assessment provided a “baseline” of plant richness and diversity of plant species for each property. Rather than determining exact area coverage for each species, we decided that a bloom count was the most quantifiable way to assess correlations between blooming species and flower visitors.

The second vegetation assessment method was a total bloom count. We recorded each plant species in flower and the corresponding total number of blooms four times per year. We surveyed four times in 2013 (May, July, September and January) and three times in 2014 (May, July and September). All flowers were counted on every blooming plant within the yard. We counted individual florets as blooms; consequently, compound flowers counted for multiple blooms. We did this for all species except Asteraceae, which were sampled at the level of the capitulum as in Hicks et al. (2016). This method was used to streamline the bloom counting process, which included over 700 species. We collected data for bloom richness and bloom abundance because educational materials for gardeners recommend planting flowers to attract pollinators (UF/IFAS Florida-Friendly Landscaping Program, n.d.; Campbell et al., 2019). While flowers are primarily often planted to attract pollinators, they may also provide valuable habitat to non-pollinating insects. On a biological level, blooms can provide refugia for non-flower visitors, or draw in pollinators for predatory insects to consume (Mahr, 2014). For this reason, we included bloom counts in our models for both flower visiting insects and for total insects.

We identified all plants to the lowest taxonomic level possible, including hybrid species. Field technicians were provided a simplified plant identification guide. This guide consisted of full color plant pictures along with species names and was developed in consultation with the primary investigator. When a confirmed identification was not possible in the field, we photographed the plant and later verified it in the laboratory. Lab identifications were made using Wunderlin’s Vascular Plants of Florida and identification by the primary investigator.

Following taxonomic identification, we subsequently categorized each plant as “Florida Friendly,” “Native,” or “Non-Native” (including native/non-native variants within a genus). “Florida Friendly” status was based on the FFL Program. We used the UF/IFAS searchable database of Florida Friendly plants to make this designation. It should also be noted that the Florida Friendly designation includes both native and non-native plants. Plants that were not considered “Florida Friendly” were designated as either Florida native or non-native using the Guide to the Vascular Plants of Florida, 3rd Edition by Wunderlin (2003) and the USDA PLANTS database (National Plant Data Team, 2020).

Insect survey protocol

To sample insect abundance and diversity, we deployed pan traps twice per month from March to December in each of the two project years. Exact dates varied among properties due to inclement weather and scheduling factors.

The pan traps each consisted of one 0.35 L white, blue, red, and yellow plastic bowl. Two traps were deployed per property per insect survey. We placed the first trap in the front yard and the second in the backyard at ground level in open areas. Trap placement was random. We mixed 15 ml of unscented, uncolored dish detergent per 3.81 L of water and filled each bowl to capacity with the soapy water solution.

Following deployment, all pan traps remained in the field for 24 h before being collected. During collection, we poured each individual bowl into an 8–12 cup coffee filter set inside a stainless-steel mesh strainer to remove the insect specimens. We placed samples from each bowl in individual sealed plastic bags. All bags were labeled with the property address, date of collection and bowl color. We subsequently placed all resulting samples in a −28 °C freezer for later processing and taxonomic identification. To prepare a sample for insect identification, we removed each bag from the freezer and allowed it to thaw. We gently washed all insect specimens off of the coffee filter and into a labeled vial using a squeeze bottle of ethanol. Once complete, we topped off the vial with additional ethanol.

We used a Leica S6D microscope to identify insects. We identified insects to a family level using Borror and DeLong’s Introduction to the Study of Insects and other taxonomic keys (Arnett & Thomas, 2000; Arnett et al., 2002; Goulet & Hubert, 1993; Iowa State University Department of Entomology, 2020; McAlpine, 1987; Triplehorn, Johnson & Borror, 2005). Due to time constraints, we did not identify insects to a narrower taxonomic level across both study years. We stored our samples with the Department of Entomology and Nematology at the University of Florida.

We subset our insect data to include a ‘flower visiting insect’ category. Floral visitors are “organisms that visit flowers for nectar or pollen but may or may not pollinate certain plant species” (Campbell, Hanula & Waldrop, 2007). We used literature citations to define insects as a flower visitor or non-flower visitor based on records of adult behavior for either all or a portion of a family (Evans, 2014; Iowa State University Department of Entomology, 2020; Newton, 1997; Smithsonian National Museum of Natural History, n.d.; Willemstein, 1987). All bees and lepidopterans were included as flower visitors (except for bee and lepidopteran families without functional mouthparts).

Analyses

We used R version 3.6.3 for all analyses (R Core Team, 2020). We compared plant survey data to insect survey data. Plant surveys were always conducted within 14 days of an insect survey.

We created general linear mixed models for insect abundance, insect richness, flower-visiting insect abundance and flower-visiting insect richness using the lme function in R. We square root transformed insect abundance and flower-visiting insect abundance in order to better meet the model assumptions. We accounted for differences between our 34 sites by adding site name and sampling date as random effects.

Our candidate models’ explanatory variables included: plant richness (total richness as well as Florida Friendly, native and non-native plant richness), bloom abundance (total abundance, Florida Friendly, native and non-native bloom abundance), bloom evenness, bloom richness (total as well as Florida Friendly, native and non-native bloom richness), site acreage, distance to the closest park, and the log of the acreage of the closest park. We calculated evenness as Evar (Smith & Wilson, 1996). Evar ranges from 0 to 1, where 0 is minimum evenness and 1 is maximum evenness. We checked that our biological explanatory variables (plant richness, bloom abundance) were not correlated to site size, distance to park, or size of nearest greenspace using simple linear regression. We checked that outliers were not driving bloom patterns by removing days with unusually high bloom counts and comparing the results with data that did not have outliers removed. The outliers had no significant effect. Similarly, removing rows of data where evenness = 1.0 (which is unusual and can also mean a complete absence of blooms) did not change our findings.

We created a list of a priori candidate models with different combinations of potential explanatory variables and then ranked and compared them using the aictab function from the AICcmodavg package.

We generated graphs using the ggplot2 package. To assess the relative strength of different variables, we used the Stats version 6.3.6 package to calculate Pearson’s correlation coefficient for each factor in the models (Majewska & Altizer, 2020). In cases when models contained interactions, we graphed the interactions in 3-D using the rms, lattice and Hmisc packages as in Gezon et al. (2018).

Results

General characteristics of sampled data

Of the 34 sites selected, average site distance to green space was 1,086.9 m, with a range of 0.02–3,039.13 m and median of 1,017.3 m. The log acreage of nearby greenspaces varied from 0.30 to 1.76 ha (mean = 0.67 ha, median = 0.40 ha), which translates to 0.0000059 to 285.18 ha (mean = 0.018 ha, median = 0.0001 ha), while the sites selected ranged from 0.08 to 0.33 ha (mean acreage = 0.15 ha, median = 0.14 acres).

After 21 months of sampling, we had a total of 247 insect surveys paired with the closest plant survey. Within this dataset, we identified 774 species of plants. Plants were somewhat closely distributed among native, non-native, and Florida Friendly species at 36.7%, 38.3% and 25.0%, respectively (the percentage is slightly less than 100% due to a small number of unlabeled plants). Average plant richness was 29.22 plants per garden per survey (min = 1, max = 211, median = 20). We counted 485,827 blooms. Of these 485,827 blooms, 12 species made up 45% percent of blooms counted (Fig. 1). Blooms varied by sampling session, with a median of 883.0 blooms, an average of 1,966.9 blooms and a range of 0 to 21,632 blooms. Bloom evenness averaged 0.24, with a median of 0.21 and range from 0.04 to 1.00.

Figure 1 Cumulative bloom count by most common plant species.

The cumulative bloom count for each plant throughout the course of the study. The top dozen species of plants dominated bloom counts by both number (see left-hand y-axis label) and percentage (see right-hand y-axis label).

Our insect surveys yielded 34,972 insects. On average 141.6 insects were collected from each garden site per survey (range = 2–540, median = 124). Insects represented 251 families and five orders (Fig. 2). The family “Dolichopodidae” from the insect survey dataset was superabundant, consisting of almost 50% of the insect abundance count and creating outliers in the data. However, removing it from the data did not make any difference in the top models, so we kept the family in the final analyses for total insects. Dolichopodidae were not included in analyses for flower visitors as they were not considered flower visitors.

Figure 2 Insect abundance by order.

The total summed insects counted for each order throughout the course of the study. The left-hand y-axis displays the count while the right-hand y-axis displays the percentage.

Sub-hypothesis 1: insect and flower visitor richness (number of species) and abundance will increase as the number of species of plants increases

Counterintuitively, higher plant richness was not associated with higher insect abundance or richness (Tables 1 and 2). Insect abundance and insect richness were both slightly negatively correlated to plant richness.

Table 1 Model selection statistics for the garden variables and landscape factors affecting four measures of insect community: (a) Insect Abundance, (b) Insect Richness, (c) Flower-visiting Insect Abundance, (d) Flower-visiting Insect Richness.

Response
(model)	K	d.f.	AICc	ΔAICc	wi	cwi	Bloom
Abundance	Plant
Richness	Fl. friendly plant richness	Native plant richness	Nonnative plant richness	Fl. Friendly bloom abundance	Native bloom abundance	Nonnative bloom abundance	Bloom evenness	Bloom abundance*bloom evenness	Bloom richness	
(a) Insect Abundance																		
	10	209	526.24	0.00	0.99	0.99		–								+		
	13	206	537.21	10.97	0.00	1.00	+			–								
	8	211	539.07	12.83	0.00	1.00		–										
(b) Insect Richness																		
	10	209	1616.24	0.00	0.35	0.35		–								+		
	9	210	1616.57	0.33	0.29	0.64				–	+							
	8	211	1618.52	2.28	0.11	0.75		–										
	9	210	1619.18	2.94	0.08	0.83	+	–										
	7	212	1619.62	3.38	0.06	0.90		–										
(c) Flower-visiting Insect Abundance																		
	10	209	806.33	0.00	0.68	0.68	+											
	8	211	808.08	1.74	0.28	0.96	+	–										
(d) Flower-visiting Insect Richness																		
	6		988.54	0.00	0.58	0.58	+		–									
	4		990.37	1.83	0.23	0.81	+	–										
	6		990.93	2.40	0.17	0.98		–										
Notes:

K, number of parameters in the model; d.f., degrees of freedom; AICc, Akaike weights; Delta AICc, Change in Akaike weights; wi, Akaike weights; cwi, cumulative Akaike weights; log L, log likelihood.

The models tested were general linear mixed models, with the exception of (d) Flower-visiting Insect Richness, which was a generalized linear mixed Poisson model. Abundance of Flower-visiting Insects and Insect Abundance were square root transformed. We tested seven to twelve models for each response variable. The top three models (or however many resulted in an AICc cumulative weight of 90%) for each response are listed below.

Plus signs indicate that a positive relationship was found, while negative signs indicate a negative relationship. Blank spaces indicate a lack of statistically significant relationships (p > 0.05).

Each model ended with the code, “random~1|Site.name/Date.veg.survey/Date.bowl.survey/bloom.date/Date.veg.survey/Date.bowl.survey/bloom.date, method = ML”, which added the mixed effects of site location and sample dates. The exception was the Poisson model for (d) Flower-visiting Insect Richness, whose syntax required additional parentheses and did not include sample dates due to model singularity errors.

Citation: https://besjournals.onlinelibrary.wiley.com/doi/full/10.1111/1365-2435.12803.

Table 2 Coefficients and relative coefficient importance for models analyzing the following response variables: (a) Insect Abundance, (b) Insect Richness, (c) Flower-visiting Insect Abundance, (d) Flower-visiting Insect Richness.

Response (model)	Value	Std. error	t-Value	p-Value	Pearson’s r	
(a) Insect Abundance						
(Intercept)	3.413	0.098	34.853	0.000***	–	
Plant Richness	−0.014	0.003	−4.031	0.0001***	−0.157	
Bloom Abundance	−0.000	0.000	−2.974	0.003**	0.072	
Bloom Evenness	−0.272	0.282	−0.966	0.335	−0.130	
Bloom Abundance*Bloom
Evenness	0.001	0.000	4.132	0.0001***	0.137	
						
(b) Insect Richness						
(Intercept)	17.69	0.936	18.900	0.000***	–	
Plant Richness	−0.117	0.031	−3.747	0.0002***	−0.143	
Bloom Abundance	0.000	0.000	−1.058	0.291	0.077	
Bloom Evenness	3.402	2.542	1.338	0.182	0.016	
Bloom Abundance*Bloom
Evenness	0.006	0.003	2.267	0.024*	0.101	
						
(c) Flower-visiting Insect Abundance						
(Intercept)	2.323	0.175	13.295	0.000***	–	
Bloom Abundance	0.0001	0.000	3.958	0.0001***	0.198	
Florida Friendly plant
richness	−0.085	0.032	−2.662	0.008***	−0.095	
Native plant richness	−0.025	0.030	−0.831	0.407	−0.058	
Non-native plant richness	0.046	0.027	1.691	0.092	−0.061	
						
(d) Flower-visiting Insect Richness						
(Intercept)	3.192	0.271	11.792	0.000***	–	
Bloom Abundance	0.000	0.000	4.732	0.000***	0.233	
Florida Friendly plant
richness	−0.142	0.050	−2.869	0.005***	−0.122	
Native plant richness	−0.029	0.046	−0.626	0.532	−0.062	
Non-native plant richness	0.057	0.042	1.370	0.172	−0.081	
Notes:

Std. Error, standard error; Pearson’s r = Pearson’s r (a measurement of relative importance. Higher numbers indicate greater importance).

The models tested were general linear mixed models. Abundance of Flower-visiting Insects and Insect Abundance were square root transformed. We tested seven to 12 models for each response variable. The coefficients of the top model for each response variable is listed below.

* Statistically significant, p ≤ 0.05.

** Statistically significant, p ≤ 0.01.

*** Statistically significant, p ≤ 0.001.

Sub-hypothesis 2: native plants > florida friendly plants > non-native plants

There was no clear indication that a certain designation of plant type resulted in higher insect abundance or richness (Tables 1 and 2). As seen in the ranked AICc tables in Table 1, we created models that had plant abundance and richness separated by type as well as models that had them not separated. In both cases, total plant richness was included in the models, but plant richness as divided by type was not.

Sub-hypothesis 3: insect and flower visitor richness and abundance will increase as the number of blooms increase

Blooms were positively correlated to flower visiting insect abundance (Table 2; Fig. 3). While graphs indicated a positive correlation to additional dependent variables, bloom abundance alone was not significant in many of our top models (Table 2). Instead, other models found that bloom evenness had a strong, positive modulating impact on bloom abundance, and bloom evenness*bloom abundance were positive and significant for insect abundance and richness (Figs. 4 and 5). In general, insect abundance and richness increased most when bloom abundance and bloom evenness both increased (Table 2).

Figure 3 Correlation between dependent variables and bloom abundance.

Scatterplots of the relationship between total bloom abundance (per site, per day) and (A) total insect abundance; (B) total insect richness; (C) flower visitor abundance; (D) flower visitor richness.

Figure 4 Interaction between bloom abundance and bloom eveness for y = sqrt(Insect Abundance).

On the x-axis is bloom abundance (in number of blooms) from 0 to 30,000 blooms. The bloom abundance axes were capped at 30,000 to make the graph easier to read. On the y-axis is the square root of insect abundance counted at each site per day. On the z-axis is bloom evenness, measured by Evar from 0 to 1.0, where 0.0 = no evenness and 1.0 = a perfectly even community. This graph indicates that bloom abundance alone did not have a clear correlation to insect abundance when evenness was low. However, as evenness increased, bloom abundance showed a strong and positive correlation with insect abundance.

Figure 5 Interaction term between bloom abundance and bloom evenness for y = insect richness.

On the x-axis is bloom abundance (in number of blooms) from 0 to 30,000 blooms. The bloom abundance axes were capped at 30,000 to make the graph easier to read. On the y-axis is total insect richness (the number of separate species of insects) counted at each site per day. On the z-axis is bloom evenness, measured by Evar from 0 to 1.0, where 0.0 = no evenness and 1.0 = a perfectly even community. This graph indicates that bloom abundance alone did not have a clear correlation to insect richness when evenness was low. However, as evenness increased, bloom abundance showed a strong and positive correlation with insect abundance.

Additional finding: bloom evenness affected insect abundance and richness

Bloom evenness had a strong, positive modulating effect on bloom abundance for both insect abundance and insect richness (Table 2; Figs. 4 and 5). In other words, as bloom evenness increased, the same number of blooms had a far larger positive impact on insect abundance and richness than they would have had with an uneven bloom distribution.

Non-significant factors

Plant type (native, non-native and Florida Friendly plants), distance to green space, and green space size (Table 1) proved to be unimportant, and models that included them were not the top models selected. We accounted for variability between sites by setting site location and date as random effects in all of our mixed models.

Discussion

Sub-hypothesis 1: insect and flower visitor richness (number of species) and abundance will increase as the number of species of plants increases

We hypothesized that plant richness would have a large, positive relationship to insect abundance and richness. To our surprise, higher plant richness was slightly negatively associated with total insect abundance and richness. Specifically, our models found plant richness to be significant, but with a weak, slightly negative relationship with insects (Table 2). For flower visiting insects, the top models divided plant richness among Florida Friendly plants, native plants and non-native plants, but only Florida Friendly plants were significant, and they too were slightly negatively correlated (Table 2).

Other literature notes the unclear nature of the relationship between plant richness and insect populations. Ebeling et al. (2008) mentioned that other field studies have found variable impacts from plant richness on pollinators, but their study, which involved highly controlled test plots, found a strong correlation. Studies by Majewska & Altizer (2020), Ebeling et al. (2008), Haddad et al. (2001) and Wright & Samways (1998) also found a positive relationship.

One possible explanation is that some taxa may be more affected than others. Smith et al. (2006) found that plant richness affected solitary bees and hoverflies, but not solitary wasps or bumblebees. Another potential reason is that all plants are not created equal. Certain plant species may be exceptionally valuable to insects, while other plants have a negligible effect (Ebeling et al., 2008; Frankie et al., 2005). For example, Ebeling et al. (2008) planted 60 unique plant species, but only 32 of them were visited by flower-visiting insects. Of those 32 popular plant species, a single especially attractive species got about half of all insect visits. When plant richness is maximized, perhaps the more attractive plants are diluted by less influential plants.

Because our study was based on observations rather than manipulated experimental plots, it is impossible to completely eliminate confounding factors. Important interactions would likely still appear despite these compounding factors, but moderate to slight factors might get lost. Alternately, perhaps evenness and plant abundance have a relationship. Because our comprehensive vegetation survey only noted the presence or absence of a given plant species, we were unable to collect evenness and plant abundance information for the total plant count, but it’s possible that having a number of the same species has a magnifying effect similar to what we saw with bloom evenness and abundance. If a site has 200 species of plants, but 100 of those species are a single individual, the effects of those low-abundance species may be negligible. We recommend further research on a potential interaction between plant evenness and abundance to guide plant selection in gardens.

Sub-hypothesis 2: for maximum insect abundance and richness, native plants > florida friendly plants > non-native plants

Our study did not find a clear advantage for insect family richness or abundance when planting native, Florida Friendly, or non-native plants. In some cases, our top models simply did not include the designations, while in other cases the top models included the designations, but only one (typically the Florida Friendly designation) was statistically significant. This result may seem surprising, considering the perception that native plants are the superior choice for a variety of reasons, including local adaptation, hardiness and attractiveness to native pollinators. Our findings provide a valuable finding for home gardeners: existing non-native plants can be beneficial to insects. Our findings are in keeping with the mixed results of other authors (Majweska and Altizer, 2020; Pardee & Philpott, 2014; Smith et al., 2006; Frankie et al., 2005; Haddad et al., 2001; Matteson & Langellotto, 2011; Burghardt & Tallamy, 2015).

On one hand, a study involving 36 experimental plots conducted by Salisbury et al. (2015) found that native and near-native treatments received higher pollinator visits than non-native treatments, and a small study on nectarivorous birds found that of four common plants, the two natives produced higher volumes of nectar per floral unit (French, Major & Hely, 2005). Likewise, Mathiasson & Rehan (2020) found that wild bee declines were linked to introduced plant species. On the other hand, a metastudy on gardener plant selection and maintenance choices by Majewska & Altizer (2020) found that native vs. non-native plant selection did not make a significant difference and recommended follow-up on plant selection. Similarly, a study by Staab, Pereira-Peixoto & Klein (2020) on exotic garden plants found that exotic species helped substitute as food resources for pollinators when native plants became seasonally scarce. The truth may be in the middle. A study by Frankie et al. (2005) found that only 9.5% of non-native flowers were actually attractive to bees, but those that were attractive were quite beneficial. This may be a piece to the puzzle: exotic flowers overall may not be bad, but simply less likely to provide food resources for flower visiting insects.

While the full answer is still unknown, this does not detract from the other benefits of planting native plants. A large body of literature already exists on the benefits of native plants. Planting hardy, well-suited plants minimizes landowner exposure to fertilizer and pesticides, saves time and money on watering, fertilizers and pesticides, reduces pollution from runoff, conserves water, controls weeds, reduces erosion, and creates wildlife habitat (UF/IFAS Florida-Friendly Landscaping™ Program, n.d.; Penn State Extension, 2019). From a research standpoint, more gardening best practices might be found by looking into the effects of host plants on their hosts, expanding on the work of Crowder et al. (2010) and Smith et al. (2006) on predatory and parasitic taxa among native and non-native plants, and expanding research on which plant traits make them more attractive to insects (Ebeling et al., 2008; Grindeland, Sletvold & Ims, 2005; Akter, Biella & Klecka, 2017; Ohashi & Yahara, 2002).

Sub-hypothesis 3: insect and flower visitor richness and abundance will increase as the number of blooms increase

Bloom abundance had a strong positive effect on flower-visiting insect abundance. This is in agreement with most literature (Majewska & Altizer, 2020; Ebeling et al., 2008; Pardee & Philpott, 2014; Akter, Biella & Klecka, 2017; Matteson & Langellotto, 2011). The strength of the interaction, measured by Pearson’s correlation coefficient r, was similar to Majewska & Altizer (2020), who found that flower abundance had a positive effect on insects with an effect size of 0.26 (compared to 0.296 for our study).

Additional finding: bloom evenness affected insect abundance and richness

While bloom abundance was important alone, bloom evenness proved to have a surprisingly influential role, strongly modulating the effects of bloom abundance in the top models for Insect Abundance and Insect Richness and appearing in the top three models for Flower-visiting Insect Abundance and Richness. As seen in Figs. 3–5, as bloom abundance increased, insect richness and insect abundance also increased modestly. However, when bloom abundance increased in conjunction with increased bloom evenness, insect richness and abundance increased dramatically.

Insects benefited most from a plant community that had both abundant and evenly distributed floral resources. Flower visitor richness and, even more dramatically, total insect abundance, increased most when more flowers were present with consistently represented blooming plants. It should be noted that complete evenness (Evar = 1.0) was not needed to benefit insects. Our focal properties had bloom evenness values primarily between 0.1596 and 0.2581. However, when blooms approached a near-monoculture, Evar values tended to be very low. Near-monocultures seemed to reduce the effectiveness of high bloom counts; thus it appears that some family richness is still an important consideration for landscapes.

Our results regarding evenness help fill critical knowledge gaps on the best practices for benefiting insects in urban landscapes. Very few studies on urban and suburban insect biodiversity have focused on evenness. Stavert et al. (2019) found that evenness by itself did not have a huge impact, but it indirectly affected plant reproduction by affecting the structure of plant-pollinator networks. Our own results on evenness by itself were negligible, but the interaction between evenness and abundance indicated a potentially interesting, statistically significant relationship. Additional research on the bloom abundance/evenness interaction could potentially maximize valuable ecosystem services such as pest control and pollination. Landscape architects, urban planners, and gardeners are already working to enhance forage and habitat resources for flower-visiting insects by increasing floral availability (Bellamy et al., 2017; Southon et al., 2017; Li et al., 2017; Todorova, Asakawa & Aikoh, 2004). When selecting plants, ensuring that different species of flowering plants will provide an even bloom presence could be a low-hanging fruit to increase the impact of plant species purchased.

Restoration projects would also benefit from increased investigation into the impact of community evenness. Restoration projects already emphasize diversity and native plant selection (Bischoff, Steinger & Müller-Schärer, 2010; Richards, Chambers & Ross, 1998; Stanley, Kaye & Dunwiddie, 2011). This approach has been successful, with Bischoff, Steinger & Müller-Schärer (2010) finding that high genetic diversity resulted in more productive plant populations with less risk of failing to establish. Observational studies on bloom evenness and bloom abundance could provide another metric for evaluating these projects. If an interaction is found, future restorations could maximize impact by prioritizing not just the total number of blooming plant species introduced, but also ensuring that the numbers planted provide an even spread of available resources. Furthermore, insects are seldom considered with habitat restoration projects and methods that include measuring evenness could assist in combatting insect and pollinator declines.

On the study location

Our study helps fill a literature gap on urban and suburban biodiversity in the Southeast United States. Most studies on urban garden habitat features and insects have been in mild to cold mid-latitude climates (Majewska & Altizer, 2020; Ebeling et al., 2008; Frankie et al., 2005; Matteson & Langellotto, 2011; Pardee & Philpott, 2014; Philpott et al., 2019; Smith et al., 2006). Our study took place in Florida, which has a humid subtropical climate (Cfa) (Kottek et al., 2006). The differences in seasonality may be significant. Most temperate climates have distinct seasons, and many plants and animals go into diapause or die off each winter (Powell & Logan, 2005). By contrast, our study site was primarily affected by two periods of growth dictated by rainy and dry seasons. This difference in seasonality resulted in differences between bloom periods and insect activity (Wolda, 1988; Lechowicz, 1995; Shimadzu et al., 2013). This in turn may have affected plant availability, resulting in a larger plant pool for homeowners to select.

Insignificant factors

Contrary to our expectations, distance to green space and the size of the nearest green space did not influence the population of flower-visiting insects. Majewska & Altizer (2020) found that many studies identified a correlation between distance to various habitats and green space, but studies have also found that that landscape level factors tended to have weaker associations with insects than within-garden features (Majewska & Altizer, 2020; Philpott et al., 2019; Pardee & Philpott, 2014; Williams & Winfree, 2013). For example, Majewska & Altizer (2020) and Philpott et al. (2019) found that landscape level factors (such as distance to agriculture, distance to water, or distance to forest) were relevant, but tended to have weaker associations with pollinators than within-garden features. In other words, external factors beyond one’s control, such as the size of the nearest park, are less important than the choices one makes with plant selection.

We also found that site size did not influence insects. However, as we selected suburban lots within a set range of sizes these conclusions are quite limited. By contrast, Majewska & Altizer (2020) found garden size was important for insects. This difference in findings reveals an opportunity for future research to explore larger datasets with a range of yard sizes. Yet, our findings provide encouraging support for the impact homeowners can have with the factors they have control over in their yards, as yard size is less flexible or easy to change.

Limitations

There were some limitations to our study regarding plant abundance, timing, trap types and species identifications. We did not include plant species abundance because our baseline vegetation surveys only recorded presence/absence of a given species and our bloom surveys did not include plant abundance counts. Thus, our vegetation surveys yielded only total plant richness and total bloom abundance. Ideally, we would have had time to also include data collection on plant abundance.

We were also limited by scheduling logistics: ideally, the insect and vegetation surveys should be collected on the same day. However, we conducted our insect and vegetation surveys on different days and then matched our insect surveys to the closest vegetation sampling day. Since most of the plants in our study bloomed for more than 2 weeks (UF/IFAS Florida-Friendly Landscaping Program, n.d.), we feel confident that we still got an accurate assessment of what plants were blooming around when we collected insects, but same-day surveys would have been preferrable.

We also only used bowl traps for insect surveys due to the use of homeowners’ private properties. These were deemed sufficient for the study since bowl traps are a widely known, widely recommended, and effective method for sampling flower visitors (Grundel et al., 2011; O’Connor et al., 2019; Westphal et al., 2008). Nonetheless, we may have been able to capture more insects with a wider array of techniques, such as malaise traps or sweep netting. We also did not account for the following variables: bare ground, water sources, unmowed areas, canopy cover, and pesticide applications.

Another limitation that would have added to the study was identifying insects to species level. Due to the volume of insects collected and labor available, we only identified insects to the family level across both years. This allowed us to identify flower visiting insects and general trends across families and orders. However, species richness is the more commonly used metric, and we were not able to determine which flower visiting insects were native, an important sub-classification. Native pollinators have experienced severe declines over the past several decades, and there is great interest (among both the public as well as scientific communities) and value in finding which variables will most help native pollinators (Potts et al., 2010; Forister, Pelton & Black, 2019; Van Klink et al., 2020; Mathiasson & Rehan, 2020). Future studies of plant-pollinator interactions could benefit by identifying insect flower visitors to a lower taxonomic level.

Conclusions

Insect populations are declining globally (Cardoso et al., 2020; Van Klink et al., 2020). We conducted this study with the goal of increasing our understanding of which suburban landscape factors contribute most to attracting pollinator and insect populations in order to help inform plant choice and related landscaping decisions.

We found that homeowners can enhance the insect richness of their backyard habitats by planting more blooming plants with an even selection of species. More blooms were clearly associated with more flower visitors, and the evenness of the blooms significantly modulated the effectiveness of bloom abundance. Planting even, abundant flowering plants of diverse species will have a much greater impact on attracting insects than planting the same number of just one species. Very few studies on urban insect biodiversity have focused on evenness, so finding that the two were related expands the literature on this topic. Our results have the potential to be applied more broadly to other landscapes within the built environment by landscape architects, urban planners, and land managers.

Native, non-native and Florida Friendly plant designations were not important in this study. Although the best fit models found no clear distinction between these three types of plants, other benefits of native landscaping remain substantial. Instead, gardeners can conclude that their existing non-native plants may still provide vital resources to flower visiting insects.

Our study highlighted opportunities for additional research. Our study was not focused on a scale to enable comparisons between individual plant taxa. Future studies might result in flower abundance guides to help decision makers find the ideal plant abundance and distribution. Likewise, both our findings and literature on plant richness were unclear. An interaction term between plant abundance and evenness may exist that explains the variable effects of plant richness.

When a suburban homeowner sets out to augment or landscape their yard there are many limiting factors they must consider, as well as factors out of their control. We found that two factors under their control-choosing flowering plants and planting even numbers of these plants-were more influential than other external factors such as nearby greenspaces. Specifically, we found that when bloom abundance and evenness increased simultaneously, there were positive, non-additive effects on flower-visiting insects. The results of our study show that homeowners can make simple changes to their landscaping decisions that will dramatically increase the impact on flower-visiting insects without an increase in cost or labor.

Supplemental Information

Supplemental Information 1 Bloom data.

Total count per day of blooming flowers at each site.

Click here for additional data file.

Supplemental Information 2 Vegetation survey data.

Total count per day of number of species of plants at each site.

Click here for additional data file.

Supplemental Information 3 Combined vegetation and insect data.

Data of combined vegetation and insect survey data. Data was paired by closest date. Variables included site name, site acreage, distance to nearest greenspace (or ‘park’ in the file), the log of the nearest park acreage, park acreage, the dates of the vegetation survey and insect bowl survey, whether the survey was a baseline survey that included all vegetation (non-basline surveys did not count non-blooming plants), plant richness, Florida-friendly plant richness (Ff.richness), native plant richness (Native.richness), non-native plant richness (Non-native.richness), location, abundance of total insects (Abundance), abundance of total insects sans dolichopodidae (Abundance.sans.dolichopodidae), insect species richness (Richness), insect evenness (evenness), lepidopteran abundance (Abundance.leps), lepidopteran richness (Richness.leps), lepidopteran evenness (lep.evenness), flower visiting insect abundance (Abundance.flower.visitors), richness of flower visiting insect species (Richness.flower.visitors), flower visiting insect evenness (flower.evenness), date of the bloom survey (bloom.date), number of blooms counted (Bloom.abundance), Florida-friendly bloom abundance (Ff.friendly.bloom.abundance), native bloom abundance (Native.bloom.abundance), non-native bloom abundance (Nonnative.bloomabundance), the richness of blooming plant species (bloom.richness), Florida-friendly blooming plant species richness (Ff.bloom.richness), native bloom richness (native.bloom.richness), non-native bloom richness (nonnative.bloom.richness), and bloom evenness (bloom.evenness). Pairs of vegetation/insect survey data which were conducted more than 14 days apart (the day.difference was greater than +/- 14) were removed. The abundance of insects with and without the dolichopodidae family were considered. We compared the number of blooms counted (Bloom.abundance) with two variants: Bloom.abundance.30000max and Bloom.abundance.sans.outliers. With Bloom.abundance.30000max, we capped bloom count at 30,000 blooms to reduce the effect of extreme outliers. With Bloom.abundance.sans.outliers, we removed datapoints where bloom count exceeded 30,000.

Click here for additional data file.

Supplemental Information 4 Raw data from insect bowls.

Raw data from insect bowl surveys. Relevant categories included date, location, order, and family. Species of insect were recorded for some bowls surveys, but not others, so we did not include this information in our article.

Click here for additional data file.

Supplemental Information 5 List of flower-visiting insect families.

Click here for additional data file.

Supplemental Information 6 Code.

Click here for additional data file.

We are grateful to the Florida Wildflower foundation, whose generous grant supported this study and inspired our topic. We thank Sandy Koi for her assistance in recruiting and coordinating study participant volunteers, and Joshua Campbell and Chase Kimmel for compiling a comprehensive literature-based reference for categorizing insects as flower visitors. We would additionally like to thank the Florida Wildflower Foundation, Florida Association of Native Nurseries, Florida Native Plant Society, Sierra Club and the Florida-Friendly Landscaping™ Program for helping advertise the request for research program homeowner volunteers, and the University of Florida’s Office of the Vice President and General Counsel for finalizing the Homeowner Permission to Access Property agreement. This research would not be possible without them. We would also like to thank the Conservation Team at Disney’s Animal Kingdom® and the Braatz family for their support throughout Elizabeth Braatz’s graduate program.

Additional Information and Declarations

Competing Interests

Author Contributions

Field Study Permissions

Data Availability

Zak Gezon, Lily Maynard, and Elizabeth Braatz are employed by Disney’s Animals, Science and Environment. Lily Maynard has worked for Project Dragonfly within the past 6 months. Jaret Daniels has been a reviewer for PeerJ in the past. Geena M. Hill is employed by Environmental Consulting and Design, Inc.

Elizabeth Y. Braatz analyzed the data, prepared figures and/or tables, authored or reviewed drafts of the paper, and approved the final draft.

Zachariah J. Gezon analyzed the data, prepared figures and/or tables, authored or reviewed drafts of the paper, and approved the final draft.

Kristin Rossetti performed the experiments, authored or reviewed drafts of the paper, performed all plant taxonomic identifications, and approved the final draft.

Lily T. Maynard analyzed the data, authored or reviewed drafts of the paper, and approved the final draft.

Jonathan S. Bremer performed the experiments, authored or reviewed drafts of the paper, performed all insect taxonomic identifications, and approved the final draft.

Geena M. Hill performed the experiments, authored or reviewed drafts of the paper, performed all plant taxonomic identifications, and approved the final draft.

Marissa A. Streifel performed the experiments, authored or reviewed drafts of the paper, coordinated project planning and logistics, including securing homeowner participation; performed all plant taxonomic identifications, and approved the final draft.

Jaret C. Daniels conceived and designed the experiments, authored or reviewed drafts of the paper, coordinated project planning and logistics, including securing homeowner participation, and approved the final draft.

The following information was supplied relating to field study approvals (i.e., approving body and any reference numbers):

Permission was given by landowners.

The following information was supplied regarding data availability:

Raw data are available in the Supplemental Files.

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
