# Peer review of "Bloom evenness modulates the influence of bloom abundance on insect community structure in suburban gardens"

_PeerJ, doi:10.7717/peerj.11132_

## Round 0.1 · original submission · Major Revisions

Thank you for submitting your MS to PeerJ. While this seems to be an interesting and relevant study, both reviewers have substantial concerns about aspects of the paper.

I agree with Reviewer 1 that it is not clear if you are conducting your statistics for 34 sites, or for traps within sites. The latter would be pseudoreplication and would require reworking. The former would be find, but then the M&M section needs to be much more clear. In general, as noted by Reviewer 1, the Methods section needs more clarity.

I also agree with Reviewer 2 that hypothesis statements are generally lacking. I can see hints of such statements in your objectives section (lines 110 - 113). Perhaps reworking that section into a number of hypotheses will also help you to restructure the remainder of the paper – methods, results and discussion – around those hypotheses. That would go a long way towards improving the overall flow, etc.

On that last point, reviewer 2 writes:

"The discussion/conclusions section is also very scattered, it seems like there was an attempt to fit the results in many broader contexts (agriculture, restoration, urban planning) and a real discussion of the results at hand was lost in this mix."

Hypothesis-based organization could help with critiques like these.

Your discussion of "Florida Friendly" is also interesting and does add value to the paper. Reviewer 2 makes note if this in their third paragraph. Please have a look at that and use the suggestion to increase the value of this section.

Both reviewers have given you a lot of ways to improve this manuscript, and I will be asking them to re-review it in the next round following receipt of your response and revisions.

Reviewer 1 ·

Basic reporting

Figures 4-6 are hard to interpret in solid black. I can't tell if there is a concavity, for example.

Line 112-113: maybe define these in the intro?

In looking over the supplemental data, I noticed that sites are coded with a name and a number. If the name used is the homeowner's last name this might violate any anonymity agreements, doubly so if the number after each name is a street address. This comment would also apply to location data (e.g. specific lat and longs, if used).

Experimental design

I have several general questions about the study and analyses

-How typical are the sites you selected? Do they represent a random sample of homeowners or is this from one specific neighbourhood? One specific demographic group?

-It is unclear to me how you treated 'sites' within your model. While you do have many observations and samples, you have them for only 34 sites. This needs to be accounted for as the variables are not independent of the site.

-Similar to the point above, it is unclear to me how you treated time in your model. You have multi-year data but don't seem to discuss or correct for variability among years.

-While I can see the model in your supplemental, it would be good to outline it more in the methods. Please state in the methods what kind of regression models you used.

Validity of the findings

Please see my points on design, above. My main concern is potential pseudo-replication when analyzing the data (i.e. how did you treat 'sites' in your study).

Additional comments

Thanks for the paper. I think it is a worthwhile contribution to the field. I do, however, have some points above that I would like addressed.

In addition to the points on the design above I have one additional question. You discuss native plants throughout but not native pollinators. What fraction of your sample represents native pollinators and do you find native pollinators and non-native pollinators have significantly different responses to the variables you've measured? If you're not measuring this (or it is too challenging given time constraints, etc.) you should absolutely discuss this in the paper.

Reviewer 2 ·

Basic reporting

The objectives of the study were to assess how the community of suburban flower visitors are affected by traits of the plant community, namely; plant species richness, bloom abundance and evenness. The authors also assessed how the greenspace area and proximity affected flower-visitors. Although these objectives are outlined as such in the manuscript, the authors also subsequently investigate the influence of the flowering plant community on insects in general. This approach is not well-justified in the text and as the results are presented, it is very difficult to understand where the authors are testing the influence of the plant community on insects in general or flower-visitors, and the rationale for each.

There are quite a few other weaknesses in the paper as well. Namely, in the framing of the work in a broader field, the lack of details in the methods, the figures and tables, and the overall presentation of results. The introduction is very scattered and does not situate the study at hand within a particular field of work. I was left wondering whether the context of this study was in biodiversity and/or ecosystem functions and services, or conservation, or applied ecology. The authors have collected a very impressive dataset, and I appreciate the effort that is involved in attaining such a large dataset of insect and plant species but the analysis was lacking depth, and the design of the study was not clearly described. Was this a ‘natural experiment’ or a biodiversity survey? Were there hypotheses to be tested? A hypothesis regarding species richness was mentioned in the discussion (line 436) but not in the introduction.

With a more in-depth analysis of the data and a sound justification for the experimental design, this may be an important contribution to knowledge of subtropical urban insect communities, due to the sheer volume of data collected. However, further taxonomic work may be needed to identify the specimens to lower level, as literature that is focused on flower visitors is usually at least to the genus level, but I realize this level of taxonomic work is logistically impossible for many studies. The most interesting result was that the Florida Friendly designation did not affect insect (family) richness, which should be a relevant finding to city planners and those concerned with biodiversity conservation in cities, however, species richness is often the more desired metric.

The figures are one the weakest points of this manuscript. They are not appropriately described or labeled and are not of sufficient resolution. Their relevance to the results is questionable. See Specific Comments file.

The manuscript is also clearly written with few grammatical errors, but there are some sections where clarity could be improved in the writing (see specific comments).

Experimental design

Overall, the experimental design and the methods are not described well enough to determine the quality of the data collection, to reproduce the data collection, and there are several issues with the statistical analyses (See Specific comments file).

The research question is not clearly defined, and as such, the manuscript is quite scattered. The study claimed to be designed to investigate the link between flowering plants and flower-visiting insects (lines 109-113), but much of the paper is focused on the insect community overall, with specimens identified to the family level only. I struggled to see rationale for the experimental design and the biological relevance for some of the analyses conducted. Namely bloom abundance on the entire insect community, the authors mention a connection to pollinators but the criteria for categorizing an insect as a flower visitor is not well-described. What is the biological relevance for testing the influence of bloom abundance and evenness on the entire insect community? This should be outlined in the introduction with supporting references.

A priori hypothesis design is an important step in an ecological study, and here it seems to have been skipped. Unless this is strictly an insect diversity survey and the authors did not intend to test any hypotheses? Because without a priori hypotheses, set within a strong evidence base, it is difficult to have full confidence in the experimental design and the results.

Validity of the findings

The lack of justification of the experimental design, and the lack of detail in the field methods and the statistical methods also make it difficult to have confidence in the findings.

The linear models should be constructed using knowledge of the system so that only predictors likely to be pertinent to the investigation are tested - here this would be those variables expected to affect flower visitor richness and abundance. There are several environmental and habitat variables that are known to affect insect abundance, especially that of pollinators, such as temperature, season, and amount of impervious surface (in urban areas). At minimum the effect of sampling date and site location should have been included as random variables in the models. See Harrison et al. 2018 for a great introduction to mixed effects modelling in ecology.

The figures are one of the weakest elements of the manuscript. The resolution is poor, the axes labels are too small and uninformative, and the figures do not clearly depict the main results. See specific comments file. An ordination plot would have been useful to include in this study to visualize differences among environmental vectors.

The supplementary files need more descriptive metadata to be useful, and I cannot find a column indicating which insects were categorized as flower-visitors? Also, there are many singletons in the data, would removing singletons change the results?
Many table and figure captions need to be revised to improve clarity. The captions should be stand-alone explanations of the figure/table contents. As is, most of them are not. For example: “Table 8 Best fit model for y = Insect richness: plantR + bloomA*bloomE”
It would help to see each variable written out in the tables as well. Instead of plantR, write Plant Rrichness. Including the response variable would also improve clarity. See Theodorou et al. 2016 (Table 2) for a great example of a table displaying the results of model testing.

The discussion/conclusions section is also very scattered, it seems like there was an attempt to fit the results in many broader contexts (agriculture, restoration, urban planning) and a real discussion of the results at hand was lost in this mix.

Additional comments

No comment.

Annotated reviews are not available for download in order to protect the identity of reviewers who chose to remain anonymous.

---

## Round 0.2 · Minor Revisions

Thanks to the two reviewers for re-reviewing this manuscript.

As you can see, both suggest minor revisions. Indeed, the suggested revisions are minor, and both note that the MS is much improved.

Please pay attention to the various issues that have been highlighted. Reviewer 2 has a variety of important suggestions, and Reviewer 1 has again mentioned potential issues with participant anonymity. Please address each of these issues in a detailed response letter.

Thanks for your work on this so far.

Reviewer 1 ·

Basic reporting

No major concerns

Experimental design

No major concerns

Validity of the findings

No major concerns

Additional comments

Thank you for your re-submission. It is greatly improved over the previous version. It looks like you've clarified most of my issues from the previous round of review.

Could the authors add a line to the MS about where their collections are stored? If they can be stored with a museum collection this would be ideal so others can go back and repeat or expand on the work.

I will raise this once more because it is unclear from your response:
In Bloom Data (S1) you still have last names and addresses as ID codes. It is very easy to use this information to find your participants. From your response, it sounded as if you were obscuring this but I don't see that to be the case. If anonymity is important, then your participating homeowners are not anonymous.

Line 341: this line seems odd, I thought some Dolichopodidae were known pollinators?

Reviewer 2 ·

Basic reporting

The authors assess how suburban insect communities (with a specific focus on flower visitors) are affected by plant community diversity, site size, and distance to greenspace, in an effort to identify landscaping practices that support insect conservation. The authors have collected a very impressive dataset, and the objectives/hypotheses are now clearly outlined in the revised manuscript. Their work is better framed in a broader field of conservation and the potential applied aspects are highlighted well in the discussion section.

This is an important contribution to knowledge of subtropical urban insect communities, especially due to the sheer volume of data collected. This manuscript in much improved from the previous versions. It is clear that the authors have done quite a bit of work in revising this manuscript and almost all of my previous comments have been addressed thoroughly and adequately. The inclusion of well-outlined hypotheses has really improved the structure of this manuscript. However, the title should be revised and the figures/tables do still need some improvement. Some of the figures are still not appropriately described or labeled (see specific comments file). Overall, the manuscript is clearly written with few grammatical errors, but there are some sections where clarity could be improved in the writing. The discussion is also quite long, and would benefit from an effort to make it a bit more succinct.

Experimental design

Overall, the experimental design and the methods are described much more clearly in this version. The statistical analyses are sound, due especially to the revision of the models and the inclusion of site and date as the random effects. The authors have now included the detail that was lacking in the first version.

Validity of the findings

The authors have now adequately justified their study design and given enough detail on their data collection and analyses methods to sufficiently replicate the study, and to give the reader confidence in their findings. The results of the modelling are clearly stated in the tables, although the tables/figures do need some revision. The limitations of the study are now clearly and thoroughly outlined in the discussion.

Additional comments

See specific comments file.

[Additional comment from Reviewer 2 added by staff as requested by the Academic Editor:
"Just a note a comment by Reviewer 1, about some Dolichopodidae being pollinators – as the author may have changed their categorization of this fly family based on my previous comment. Maybe you can pass on the reference to the authors to justify their categorization as non flower-visitors. Their role as pollinators is understudied, but as a family they largely not considered flower-visitors. This assertion comes from Table 1 of Larson, B. M. H., P. G. Kevan, and D. W. Inouye. "Flies and flowers: taxonomic diversity of anthophiles and pollinators." (2001). Although there may be more recent studies, I am unaware of any."]

Annotated reviews are not available for download in order to protect the identity of reviewers who chose to remain anonymous.

---

## Round 0.3 · accepted · Accept

Thank you to the reviewers for their excellent work, and to the authors for comprehensive responses in both rounds of review. This manuscript is now ready for publication in PeerJ.